# Analysis of Environmental Activities for Developing Public Health Investments and Policies: A Comparative Study with Structure Equation and Interval Type 2 Fuzzy Hybrid Models

**DOI:** 10.3390/ijerph17061977

**Published:** 2020-03-17

**Authors:** Cuina Zhang, Ruobing Li, Yun Xia, Yixing Yuan, Hasan Dinçer, Serhat Yüksel

**Affiliations:** 1School of Technology, Harbin University, Harbin 150086, China; gxzhangcn@hrbu.edu.cn; 2Heilongjiang Province Academy of Cold Area Build Research, Harbin 150080, China; vanness565821@foxmail.com; 3School of Environment, Harbin Institute of Technology (HIT), Harbin 150006, China; alexandros2752127@foxmail.com; 4School of Business, Istanbul Medipol University, Kavacık South Campus, Beykoz, 34810 Istanbul, Turkey; hdincer@medipol.edu.tr

**Keywords:** physical activity, environment, public health, structure equation model, DEMATEL, TOPSIS, interval type 2 fuzzy sets

## Abstract

The design of elements which exert pivotal effects on leisurely physical activity (LPA) in open space is an important part of urban development. However, little research has been done about the influence and discrepancies of those elements in different types of open space. To research these issues and to guide the design of urban open space, a survey from 8 open spaces (2 curtilage, 2 neighborhood squares (NS), 2 parks, and 2 campus) is conducted and a questionnaire is administered. Simultaneous analysis of several groups (SASG) of Structure equation model (SEM) is used, and the effects and discrepancies are acquired. In addition to this situation, interval type 2 (IT2) fuzzy hybrid decision making model is proposed in the second analysis. In this framework, IT2 fuzzy decision-making trial, evaluation laboratory (DEMATEL), and IT2 fuzzy technique for order preference by similarity to ideal solution (TOPSIS) methods are used. The results show that the influence relationships between elements and LPA did exist in four groups. Another important conclusion is that there were discrepancies of influence among different space groups. Physical environment (PE) has the greatest influence on LPA in the curtilage, whereas facilities exert the most effect in NS group. Additionally, amenities only have significant impact in parks and facilities only exercise remarkable influence on duration on campus. In addition to them, it is also identified that key design elements are presented for different types of space and that design strategy is provided through 4 specific examples.

## 1. Introduction

Public health policies aim to maintain a healthy life and to protect people from diseases [1,2,3]. There are a number of practices that are necessary to increase public health [4,5,6]. Physical activity of the public plays an important role for the improvement of public health. The main reason for this is that physical inactivity can cause some important health problems, such as obesity [7]. Therefore, people living in the country need to be encouraged to be more mobile. For this purpose, there are some things people can do [8,9,10]. For example, walking or cycling for transportation will help people become more active. In addition, performing fitness exercises and participating in sports activities are other practices that can minimize physical inactivity. These types of physical activity will contribute to the health of people. There are a number of strategies that can be implemented by countries to increase this issue. One of the most important issues in this process is the creation of environments within the country to support physical activities [11].

There are many different opportunities for people to have leisurely physical activity (LPA). In this context, there can be some facilities in which citizens can join some sport activities. For instance, wide open space for dancing and paths for walking and jog can contribute LPA [12]. In addition to these issues, play equipment for kids and fitness equipment for adults are also other examples for this situation. Moreover, physical environment (PE) also plays a very key role for LPA [13]. Within this framework, the trees can be planted for shade, and fresh air and adequate trash bins can be provided to maintain sanitation. The main reason is that PE, such as noise and air, has impact on physical activities. Additionally, amenities, such as tables, retail stores, and toilets in the parks can promote LPA. Furthermore, aesthetics can also make contribution to higher LPA. For this purpose, green plants and water bodies can help people to become physically active. In addition to these factors, maintenance and safety (MS) can have a positive influence on LPA as well [14,15].

In this context, it is very important to determine which physical activity will have more impact on public health. In other words, there are many different factors that improve public health. However, it may not be possible for policy makers to invest in all of them at the same time. It is possible to talk about many different reasons for this restriction [16]. Primarily, investing in the development of physical activities is very costly. Therefore, the state institution may not have sufficient budget to invest in all factors at the same time. Therefore, it is very important to identify and focus on priority investment areas in order to develop physical activities. In this way, it will be possible both to contribute to the development of physical activities and to use the public budget effectively [17,18].

Structural equation model is one of well-known methods in order to examine the causality process between multiple variables especially in social sciences [19]. This method has been preferred for different purposes by different researchers, especially in recent years. It is thought that it will be effective in the emergence of the important issues in the analysis to be carried out in order to increase physical activities. In addition to the mentioned issue, multi-criteria decision-making methods (MCDM) are approaches used for this purpose in the literature. With methods such as analytic hierarchy process (AHP), analytic network process (ANP), and DEMATEL, it is possible to identify the most important among the different alternatives [20,21,22,23]. These methods have also been taken into consideration within the framework of fuzzy logic in recent years. Considering the mentioned issues, it will be possible to determine which physical activities affect public health more with these approaches.

The allocation and user requirements of elements differ in various types of urban open space. This situation also leads to diverse effects of elements on LPA in different spaces. However, little research has been done on discrepancies of this influence. In this paper, it is aimed to evaluate environmental activities for developing the public health investments and policies in Harbin, China. In this scope, the literature is reviewed and 6 factors that influencing the environmental activities are defined. Additionally, duration and frequency are also taken into consideration. On the other side, 4 types of open spaces are also considered. For this purpose, two different analyses are performed. First of all, an evaluation occurs by using a structural equation modeling. Moreover, IT2 fuzzy hybrid decision making model is proposed in the second analysis. In this framework, IT2 fuzzy DEMATEL and IT2 fuzzy TOPSIS methods are used.

The most important contribution of this study is performing two different analyses. With the help of this issue, a comparative analysis can be made, and this situation provides an opportunity to test the accuracy of the results. Hence, it is thought that the policy recommendations of this study can be more appropriate. In addition to this condition, structural equation modeling, IT2 fuzzy DEMATEL, and IT2 fuzzy TOPSIS methods are used together firstly in this study regarding the evaluation of environmental activities for developing the public health investments and policies. Moreover, the results of this study will be leading both policy makers and academicians. In other words, by considering the analysis results, it can be much easier to improve public health with the help of correct policies with respect to the environmental activities.

This study consists of 5 different sections. In the first part, it is stated why issues such as physical activity and public health are important. Similar studies in the literature are summarized in the second part of the study. In addition, the third part of the study provides information about the methods used in the analysis. In the fourth part of the study, the details of the two different analyzes are given. In this framework, both the structural equation model and the analysis made with IT2 fuzzy multi-criteria decision-making methods are shared. The last part includes the discussion and conclusion.

## 2. Literature Review

The issue that environmental factors are also effective on public health has been discussed in the literature by many researchers. In this process, one of the most significant topics discussed by the researchers is the physical activity. Within this framework, some researchers argued that physical activities can be improved when there is low environmental pollution. For this situation, Ogasawara et al. [12] and Finch et al. [24] identified that the prevention of hazardous waste and that minimizing environmental pollution lead to higher physical activities, and this issue also contributes to the improvement of public health. Similarly, Armand et al. [13] and Lu et al. [25] focused on the relationship between environmental pollution, economic development, and public health. It is determined that, if air and water are not polluted, it can be more possible to increase physical activities which are prominent issues for citizens to lead a healthy life.

In addition to them, some researchers in the literature also defined that green areas are very important to increase the physical activities of the citizens. For this situation, it is mainly discussed that, in the country, there should be lots of green areas for the people to perform these activities. In this framework, McLafferty and Murray [14] and Moretti et al. [26] aimed to evaluate the environmental and human health impact of road construction activities. They mainly stated that physical activities will be adversely affected if the green areas are destroyed and construction is done. Parallel to these studies, Henry and Price [15] and Pilkington et al. [27] also focused on the effects on environmental factors on the physical activities. In this study, it is identified that, if environments are not well designed, there is a risk that physical activities in this country can be lowered.

Moreover, Hunter et al. [16] and Cronk and Bartram [28] also tried to examine the environmental conditions in health care facilities in low-and middle-income countries. According to the results of the analysis, it is identified that, in order to increase public health, citizens should have areas to perform physical activity. In this context, environmental factors play a very key role for the people to join physical activities. In other words, if there are lots of green areas in the country, this situation attracts the attention of the people to do these activities. Chauvin et al. [17] and Jang et al. [29] made also a similar study for this issue. They also underlined the importance of green areas to improve physical activities in the country. In addition to these studies, Kojan et al. [19] and Morici et al. [30] also made an analysis to understand the relationship between environmental issues and physical activities. They reached the conclusion that measures should be taken to reduce environmental pollution in order to improve physical activities. Hence, it can be much easier to implement an effective public health policy.

In the literature, some researchers also identified the main advantages of the physical activities in improving public health. It is observed that the public can prevent many diseases by doing physical activities. For example, Dudley et al. [31] made an evaluation regarding the physical literacy policy in public health. In this study, it is mainly concluded that physical activities play a very significant role to improve public health. Therefore, it has been stated that the public should have environments that can perform physical activity. Additionally, White et al. [32] also aimed to examine physical activity in natural environments to have better health condition. In this context, it is recommended to put physical activity equipment in places such as the park that everyone can reach.

In addition, some studies in the literature determined that the creation of areas such as walking trails, where the public will engage in physical activity, will also contribute to the development of this issue. Hulteen et al. [33] aimed to understand how global participation in sport and leisure-time physical activities can be increased. They argued that necessary areas should be created for the people to make physical activities. Parallel to this study, Khan et al. [34] also focused on the patterns of physical activity in adolescents in Dhaka city of Bangladesh. They mainly stated that, for a more effective public health policy, the public should be ensured to participate in physical activities. Kobau et al. [35] and Saunders et al. [36] also determined that, as a result of physical activities, some diseases will decrease, and this will contribute to the improvement of public health.

As a result of the literature review, it was determined that many different analyzes were made on the subject of public health. In the studies, issues such as the importance of public health, how it can be increased, and what it has a positive effect on were examined. In these studies, it is seen that regression analysis and survey methods are generally preferred. However, it is thought that more specific suggestions are needed to increase public health and that studies are needed. This study focused on physical activity to increase public health. In this framework, a study has been conducted on what kinds of physical activity opportunities should be created. Therefore, it is thought that this detailed study will be directed towards this need stated in the literature.

## 3. Materials and Methods

In this paper, it is aimed to examine the environmental activities for developing the public health investments and policies in Harbin, China. For this purpose, two different analyses are performed. First of all, an evaluation occurs by using a structural equation modeling. Moreover, IT2 fuzzy hybrid decision-making model is proposed in the second analysis. In this framework, IT2 fuzzy DEMATEL and IT2 fuzzy TOPSIS methods are used. IT2 fuzzy sets aim to minimize uncertainty of interval type-1 fuzzy sets. For this purpose, upper and lower trapezoidal membership functions are considered [37,38]. The details of calculations of IT2 fuzzy sets are given in the Appendix A. DEMATEL approach is considered to find the significance values of different alternatives. It is also possible to make impact relationship analysis with this methodology. Thus, it is thought that DEMATEL has some advantages over similar methodologies [39]. For instance, it is possible to generate impact relation map between the criteria. In this process, initial direct-relation fuzzy matrix, normalized matrix, and total relation matrix are created so that the weights of the factors can be defined. The details of this process are also shared in the Appendix A as well. On the other side, TOPSIS methodology is also used to rank different alternatives according to their importance. This approach is mainly considered to understand which alternatives are more effective in comparison with others [40]. Its calculation process is also explained in the Appendix A.

Moreover, a comparative analysis is applied for ranking the alternatives of environmental activities to develop the most appropriate public health policies. For this purpose, 6 factors that influencing the environmental activities are defined based on the literature review, and they are presented in Table 1.

As can be seen from Table 1, 6 different environmental factors were identified in order to increase physical activity. First of all, it is possible to increase the participation of the people in physical activities with the help of some facilities. In this framework, walking track and exercise equipment contribute to achieving this goal. On the other hand, these facilities must be at an accessible location in order to increase physical activities. Otherwise, it will not be possible to sustain these physical activities. In addition, the quality of the physical environment is important in increasing physical activities. In this context, it is an important advantage to find forest areas in the living environment. In addition, the fact that the air is not dirty plays an important role in this framework. Moreover, amenities, such as tables, retail stores, and toilets in the parks can contribute the improvement of the increase physical activities. Also, aesthetics like green plants and water bodies can help people to become physically active. Additionally, people who want to join physical activities prefer to feel secure. Different environmental activities for the public health are determined to analyze the leisurely public activities. For this purpose, 4 alternatives of leisurely public activities are selected as curtilage (alternative 1), neighborhood square (alternative 2), park (alternative 3), and campus (alternative 4). The evaluations are provided by considering the environmental activities of Harbin in China.

In this study, two different analyses are performed. In the first stage, Structure Equation Model (SEM) is applied for understanding the behavioral results more accurately. This model is one of the most important statistical methods in the field of behavioral and social sciences. Simultaneous analysis of several groups (SASG) can be used to analyze whether the theoretical model proposed by the researchers is the same in different groups. It can also test the discrepancies of related paths in the same group. In this paper, the discrepancies of the relationships in 4 types of open spaces is studied by SASG. Randomly selected respondents are administered a questionnaire survey. Additionally, three pieces of information including the quality evaluation of the elements, self-rated LPA (the frequency every week and duration every time), and users’ socioeconomic background are collected. The evaluation of elements is measured by a five-point Likert scale, which use very poor, poor, average, good, and very good. Elements include 6 different domains (facilities, accessibility, physical environment (PE), amenities, aesthetics, and maintenance and safety (MS)), and 42 items involved in the questionnaire are derived from the previous studies. Furthermore, 400 questionnaires are randomly distributed in 8 urban open spaces in Harbin, including 4 space types (curtilage, NS, park, and campus). The survey recovers 322 valid questionnaires with a recovery rate of 80.5%.

In the second stage, a hybrid fuzzy decision-making model based on IT2 fuzzy sets is used for comparing the analysis results of factors and environmental activities. Three decision makers that are experts in the field of environment and public health are appointed for the linguistic evaluations for the factors and physical activities. IT2 fuzzy DEMATEL is applied for weighting the factors of environmental activities, and IT2 fuzzy TOPSIS is used for ranking the alternatives of leisurely physical activities. Thus, it is possible to understand the coherency of analysis and to develop the public health policies in detail. In this study, two different analyses are performed. With respect to the structure equation model results, a survey study is made with 322 different respondents. The details of the survey questions and these people are given on the Appendix A ( Table A1).

## 4. Results

In this study, two different analyses are performed. In the first aspect, an evaluation is made by using a structural equation modeling. After that, IT2 fuzzy hybrid decision making model is proposed in the second analysis. In this section of the study, two different analysis results are given.

### 4.1. Analysis Results of Structural Equation Method

The reliability test is carried out by Cronbach’s α coefficients, and it is calculated as the whole scale and each group. All α coefficients are greater to 0.8 (0.809~0.957). It means that the reliability of the whole scale and each group are all significant. The construct validity of the questionnaire is analyzed by exploratory factor analysis. Kaiser–Meyer–Olkin measure and Bartlett’s test of sphericity (KMO = 0.928, Sig = 0.000) indicates that the validity of the questionnaire is suitable for further analysis. According to the past study achievements, the research made an assumption of the influence of facilities (I1), accessibility (I2), PE (I3), amenities (I4), aesthetics (I5), and MS (I6) on the frequency (F) and duration (D) of LPA. It also considered the influence among elements, and the final influencing model is established as shown in Figure 1. All the sample dates are introduced into the model, and nonsignificant routes are removed; the final influencing model (full model) is obtained as shown in Figure 2 after repeated fitting. All the fitting indexes in the final model are shown in Table A2. In view of the complexity of the model, three simplicity indexes, Parsimonious Normed Fit Index (PNFI), Parsimony Goodness of Fit Index (PGFI) and Parsimony Comparative Fit Index (PCFI), have a slight difference from the fitting standard, with the rest of the indexes up to the standard. As a result, it can be said that the model fitting is good and meets the requirements.

It is known from the final influencing model that facilities have a direct influence on the duration and frequency of LPA. On the other side, it is also defined that accessibility and PE directly influence the frequency and duration, respectively. Furthermore, amenities and MS exert an indirect influence on frequency by means of impacting other elements. Similarly, aesthetics also has an indirect influence on frequency and duration. SASG is performed with space types as the moderator variable on the basis of the above final model. The discrepancies of the relationship between elements and LPA in different space groups is analyzed, and the results are shown in Table A3. The PE of the curtilage exerts a significant influence on LPA. Additionally, the facilities, accessibility, PE, and MS of NS all have notable influence. In addition to them, facilities, accessibility, amenities, aesthetics, and MS all significantly affect LPA in the park. Moreover, facilities, aesthetics, and MS exercise remarkably impact campus LPA. The effect among the elements also exists in 4 groups, as shown in Table A4.

The significant discrepancies of relevant paths within groups are examined by SASG; the results are shown at Table 2, combined with Table A3 and Table A4. The facilities and PE of the curtilage have sharply different influences on LPA, and only the latter exerts obvious influence. MS and amenities have significantly different impacts on the PE, and only the latter exerts drastic influence. Facilities and accessibility of NS have drastically different effects on frequency; the former is greater than the latter. In the park group, facilities and accessibility have great differences on frequency, with the former having notable influence and the latter having nonsignificant influence. The facilities and PE of campus have different influences on duration, with the former having notable influence and the latter having nonsignificant influence; MS and aesthetics have enormously different influences on the PE, with the former having nonsignificant influence and the latter having notable influence.

Significant discrepancies among groups can be seen from Table A3, Table A4 and Table A5. The facilities in the NS and parks groups have significant influence on frequency, with the former exerting greater influence than the latter. Aesthetics in the campus and other three groups has tremendously different impacts on the PE, with the former having remarkable influence and the latter three exerting insignificant influence. MS in NS and the other three groups has sharply different effects on amenities, with the influence in the NS greater than that in campus with the influence of the other two being insignificant. Additionally, facilities in NS and parks have different influences on frequency, with that of the former greater than the latter. From the above analysis, the special elements are found which should be considered in the design of urban open space.

### 4.2. Analysis Results of Fuzzy MCDM Approaches

On the other side, regarding the interval type 2 fuzzy hybrid model results, firstly, 6 criteria of environmental activities as well as duration and frequency are analyzed to employ the relation matrix among the factors. The experts give their opinions by considering the linguistic scales as seen in Table A6 [41]. On the other side, the linguistic evaluation results for the relation matrix by the decision makers are presented in Table A7. Linguistic scales are adapted to the trapezoidal fuzzy number for analyzing the relative importance and directions among the factors. For that, fuzzy direct relation matrix is constructed, and the results are given in Table A8. The averaged values of decision makers are considered to compute the final fuzzy direction matrix. In the following process, the normalized and defuzzified values are computed to construct the total relation matrix and impact-relation map of factors. The total relation matrix as well as the directions and the weights of factors are presented in Table 3.

According to the results, I1 (facilities) is the most important factor while I2 (accessibility) has the weakest priorities in the factor set. The weighting results are obtained by normalizing the values of r + y, and the values of r − y show the influence directions among the factors. Accordingly, Figure 2 illustrates the relation map of factors.

The average value of total relation matrix is defined as a threshold, and higher values than threshold are selected as there is an influence between the factors. Most factors have an impact on duration and frequency. I2 (accessibility) has the least directions on the other factors as maintenance and safety (I6) influences the most. It is understood that the directions among the factors are coherent when it is compared with the results of the structure equation model. However, the alternatives of physical activity are ranked with TOPSIS based on interval type 2 fuzzy sets. For this purpose, the decision makers provide their linguistic evaluations by considering the evaluation scales stated in Table A9 [42]. Linguistic evaluations of alternatives are collected by the decision makers, and the results are represented in Table A10. Linguistic evaluations are converted into the fuzzy numbers, and the averaged values are presented as decision matrix in Table A11. After this step, defuzzified values of decision matrix are computed, and the weighted matrix is constructed by considering the results of IT2 FDEMATEL. The results are given in Table A12. In the following process, the values of D+ and D− as well as the closeness coefficient are calculated to rank the alternatives. The values are presented in Table 4.

The values of CCi are ranked by decreasing order. According to the results, park (alternative 3) is the best alternative among the environmental activities while neighborhood squares is the worst alternative of physical activities. However, the results of closeness coefficient by the alternatives are similar, so each alternative of environmental activity is equally important for public health investments. The government could develop the investments of physical activities by considering park (alternative 3), curtilage (alternative 1), campus (alternative 4), and neighborhood squares (alternative 2).

## 5. Policy Recommendations for Public Environment

### 5.1. Curtilage

In the group curtilage, the PE has the greatest influence on duration in the four groups. It is because such a type of space is mainly distributed in old neighborhood communities suffering poor maintenance and bad sanitation. Other elements do not have significant influence possibly because of insufficient layout or non-configuration. Moreover, compared with other groups, amenities have outstanding influence on the PE because of the influence of such amenities as trash cans on the air quality of PE. Curtilage, which mainly exists in the old community, is not renovated for lack of fund. The most urgent task now of curtilage is to improve the PE and the amenities and to provide some facilities to meet the basic needs of the residents to take LPA. The following is taking the curtilage in XuanXi community as an example to elaborate design countermeasures.

This community is ropey, and most of the elements are in poor quality. The users are the community residents, with most being old people and children. To improve the “PE”, the trees are planted for shade and fresh air and adequate trash bins are provided to maintain sanitation. To afford a certain number of “facilities” to meet the basic needs of the residents, those strategies are presented: establishing the human–vehicle branch system; setting roadside parking and zebra crossing to prohibit the entry of vehicles; and providing the following facilities: wide open space for dancing, small opening for shadowboxing, a path for walking and jogging, play equipment for kids such as a swing, fitness equipment for adults, rest facilities for sitting or chess, and lighting for night and safety.

### 5.2. Neighborhood Squares

In the group of squares, “facilities”, “accessibility”, “PE”, “aesthetics”, and “MS” have influence on LPA. Compared with other groups, “facilities” have the greatest impact, indicating that perfect “facilities” in squares can promote LPA. Such a type of space is represented by the central square of the new communities, and the task of square space design is to diversify the types of “facilities” to meet the demands of people of different ages. Meanwhile, due to the influence of other elements on LPA, the design strategies including “accessibility”, “PE”, “aesthetics”, and “MS” should also be developed just like those in the square of Rui Cheng community.

This is a new community with a large open square in which there are fitness equipment, trees, pool, etc. The main users are the elderly people, children, and adolescents of community residents. To improve the existing facilities to meet the needs of multiple age users, many facilities are configured including wide open space for dancing; small opening for shadowboxing; a path for walking and jogging; fitness equipment; a court for teenagers; resting facilities such as seats, pavilions, gazebo, etc.; billboards advertising fitness, etc.; lighting for courts; and evergreen plants.

### 5.3. Park

In the group of parks, “facilities”, “accessibility”, “amenities”, “aesthetics”, and “MS” have remarkable influence on physical activities. “Facilities” exerting greater influence than “accessibility” illustrates that users pay more attention to the configuration of “facilities” in the parks. Additionally, “amenities” only have noticeable influence in this group because such “amenities” as tables and toilets in the parks can promote LPA there. Urban parks are important places for the people to take LPA and are used by people of all ages. The present task of design is to satisfy the demands of people of different ages and backgrounds, focusing on providing comprehensive fitness facilities and improving “amenities” such as “toilets”, “tables”, “retail stores”, and so on. “Aesthetics” and “MS” elements should be strengthened concurrently. Specific design strategies are provided in the following instance.

Yellow River park: The “facilities”, “amenities”, “aesthetics”, and “MS” elements are better in this park, but the “ball fields”, “accessibility”, and “toilet” are also significantly inadequate. The users’ background is complex and various, with many belonging to each age group and coming from surrounding residential areas including the high-, medium-, and low-income communities. The design objectives should meet the needs of a variety of age groups. A comprehensive and perfect sports fitness service should be provided. In the future, we should focus on improving the “amenities” such as “toilet”, “drinking”, “retail”, etc., adding the following facilities: retail store; overpass to promoting accessibility; play equipment or ground for kids; security for kids such as soft ground, etc.; park services department; toilet; warning sign; sculpture made of snow or ice; increased categories of evergreen plants; resting facilities such as seats, pavilions, gazebo, etc.; chess tables; large trees shielding the wind; parking; pedestrian entrance; zebra crossing; top-view map of park; and obstacle-free caring design.

### 5.4. Campus

In the group of campus, “facilities”, “aesthetics”, and “MS” have sharp influence and facilities have remarkable influence on “frequency” and “duration”. The importance of providing “facilities” on campus for the people can be shown. This kind of space is a supplementary space for public LPA, the design of which should consider the needs of urban residents around the campus. The primary task of design is to provide site for football, basketball, standard circular rubber lanes, and so on. The “aesthetics” and “MS” have important effects on LPA which should be reinforced.

Campus of Harbin Institute of Technology (HIT) Due to serving university students mainly, the campus lacks fitness facilities, play equipment for kids, and other elements for surrounding residents. The users cover all kinds of people including university teachers, family members of teachers, and other residents from surrounding communities containing a wide age distribution such as children, teenagers, youth, and middle-aged and old people. The design objectives are providing fitness equipment and free ball fields to the surrounding residents, perfecting auxiliary facilities to attracting more people to participate in LPA, adding the following facilities: free court or playground and runway; paved small opening for shadowboxing; a path for walking and jog; play equipment or ground for kids; fitness equipment for adults; table for placing objects; seats for rest; billboards advertising fitness; lighting for courts; pedestrian walkways; entrances; and toilets.

## 6. Conclusions

This paper tries to evaluate environmental activities for developing the public health investments and policies in Harbin, China. Within this framework, as a result of the literature review, 6 factors that influence the environmental activities are identified. Moreover, duration and frequency are also considered. Furthermore, 4 types of open spaces are also taken into consideration. In this scope, two different analyses are conducted. Firstly, an evaluation occurs by using a structural equation modelling. Additionally, IT2 fuzzy hybrid decision making model is proposed in the second analysis. Within this context, IT2 fuzzy DEMATEL and IT2 fuzzy TOPSIS methods are used.

The main limitation of this study is making an evaluation only for Harbin. Therefore, in a new study, an analysis can be conducted for other regions or countries. With the help of this issue, different recommendations can be presented. In addition to this condition, another important limitation of this study is related to the methodology. Hence, it is recommended that different methods be considered. For instance, probit, logit, and regression analyses can be conducted. Hence, it can be possible to compare the results of different studies. The main reason is that probit, logit, and regression analyses consider quantitative data whereas SEM and fuzzy MCDM approaches use the opinions of people and experts. Thus, it can be possible to make a comparative analysis. Moreover, it can also be accepted as the limitation of this study that only the concept of physical activities is evaluated. In addition to the physical activity, there are also lots of other factors which have a contribution to the public health, such as economic growth and green environment. In future studies, these factors can be taken into account.

According to the analysis results, it is determined that facilities play the most significant role to improve LPA. In addition to this situation, it is also concluded that maintenance and safety is also another important criterion to increase these activities. However, it is identified that accessibility and amenities are the least important items for this purpose. Another important conclusion of this study is that both SEM and IT2 fuzzy logic analysis results are quite coherent. Hence, it is recommended that some sport activities, such as wide-open space for dancing and a path for walking and jogging can contribute the improvement of LPA. On the other side, it is also learnt that people who want to join physical activities prefer to feel secured. Thus, governmental authorities should take necessary actions for the people to feel secure while joining these activities. In this context, it is important that sports equipment comply with quality standards. Therefore, it is necessary to make legal regulations that require the equipment to comply with these standards. In this way, people will feel safer when using sports equipment. This will contribute to the increase of LPA activities. In the literature, McLafferty and Murray [14] aimed to focus on the regional perspectives on public health in their studies. They also underlined the importance of maintenance and safety to improve these activities. Similar to this study, Henry and Price [15] also reached similar conclusions. In addition to them, Cron and Bartram [28] aimed to evaluate the environmental conditions in health care facilities. For this purpose, low- and middle-income countries are taken into account. They defined that environmental factors play a very key role for the people to join physical activities. Parallel to this study, Jang et al. [29] also determined that environmental factors play a very key role for the people to join physical activities.

SGSA of SEM indicates that the PE of the curtilage exerts significant influence on LPA; that the “facilities”, “accessibility”, “PE”, “aesthetics”, and “MS” of the NS have obvious influence; that the “facilities”, “accessibility”, “aesthetics”, and “MS” of parks boast drastic influence on LPA; and that the “facilities”, “aesthetics”, and “MS” on campus exercise remarkable influence. Discrepancies of influence exist among difference types of open spaces. PE has the greatest influence in the curtilage group. Facilities have the most obvious influence in the NS group. Amenities only have notable influence in the park group. Facilities only exert tremendous influence on duration of LPA in the campus group.

Design strategies have been made according to the analysis results and the characteristics of different spaces: Stress should be given to the design of elements such as the PE, facilities, and amenities to meet the basic needs of the residents: improving the existing facilities to meet the needs of multiple age users; adding facilities for adolescents; and strengthening environmental sanitation and amenities maintenance in NS. In parks, comprehensive and perfect sports equipment should be provided and the “amenities” such as “toilet”, “drinking”, and “retail” should be accounted for as should the accessibility, aesthetic, and MS. Attention should be paid to the demand of LPA on facilities, aesthetics, and MS of the surrounding residents in the campus space. This study hopes to provide a reference for the development of policies, management, and sustainable urban design.

## Figures and Tables

**Figure 1 ijerph-17-01977-f001:**
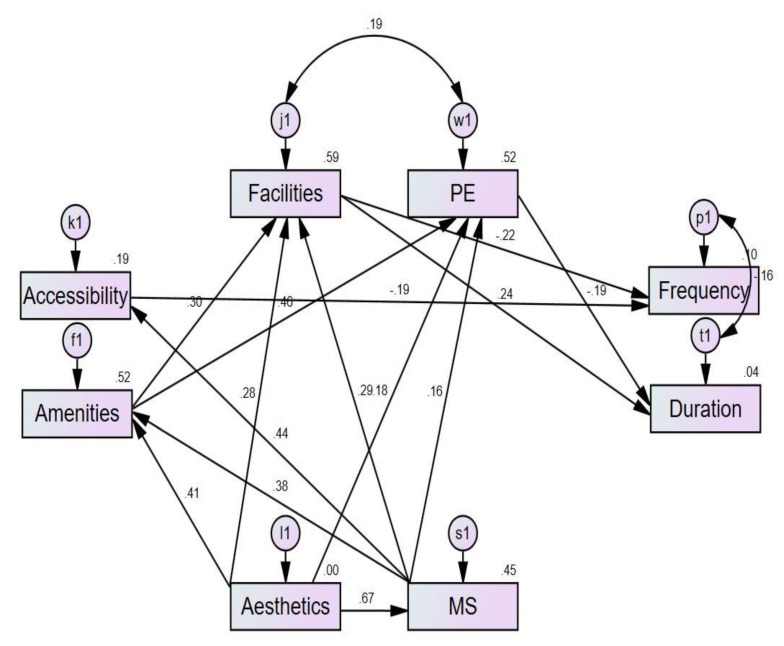
Final influencing model.

**Figure 2 ijerph-17-01977-f002:**
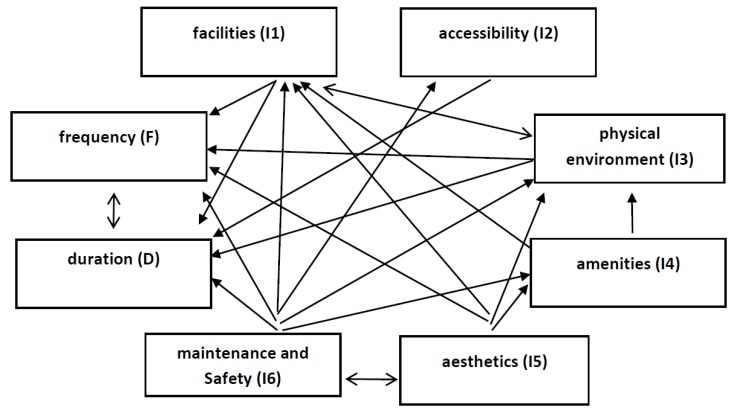
Relation map of factors.

**Table 1 ijerph-17-01977-t001:** Factors influencing the environmental activities.

Factors	Supported Literature
facilities (I_1_)	[28,29]
accessibility (I_2_)	[30,32]
physical environment (I_3_)	[33,34]
amenities (I_4_)	[29,36]
aesthetics (I_5_)	[24,25]
maintenance and safety (I_6_)	[17,18]

* I represents different criteria.

**Table 2 ijerph-17-01977-t002:** Differential critical ratio of paths in the groups.

Path	Differential Critical Ratio			
	Curtilage	NS	Park	Campus
D←I_3_, I_1_	2.137 *	1.696	1.555	1.979 *
F←I_1_, I_2_	0.222	−1.964 *	−1.968 *	−1.017
I_3_←I_4_, I_5_	−1.809	−1.115	−2.455 *	1.908
I_3_←I_4_, I_6_	−2.457 *	−1.286	−3.377 ***	−0.741
I_3_←I_5_, I_6_	0.081	−0.138	0.020	−2.986 **
I_1_←I_4_, I_5_	−0.326	0.702	−0.252	0.339
I_1_←I_4_, I_6_	−1.781	−0.353	−0.538	−0.992
I_1_←I_5_, I_6_	−1.004	−1.090	−0.161	−1.295
I_4_←I_6_, I_5_	1.065	−1.130	2.990 **	1.227
* *p* < 0.05; ** *p* < 0.01; *** *p* < 0.001				

* *p*: probability; D: duration; F: frequency; I: criteria; NS: neighborhood squares

**Table 3 ijerph-17-01977-t003:** Total relation matrix and impact-relation degrees of factors.

	I1	I2	I3	I4	I5	I6	D	F	r + y	r − y	Weights
I1	0.19	0.15	0.34	0.14	0.16	0.15	0.30	0.34	3.88	−0.31	0.172
I2	0.17	0.05	0.17	0.11	0.14	0.08	0.24	0.18	2.06	0.22	0.091
I3	0.31	0.09	0.14	0.11	0.09	0.09	0.18	0.31	3.08	−0.44	0.136
I4	0.35	0.08	0.28	0.07	0.08	0.09	0.13	0.16	2.47	0.01	0.109
I5	0.43	0.14	0.28	0.27	0.12	0.23	0.17	0.20	2.92	0.80	0.129
I6	0.45	0.31	0.33	0.32	0.30	0.13	0.21	0.23	3.19	1.36	0.141
D	0.06	0.03	0.06	0.06	0.04	0.04	0.07	0.21	2.14	−1.01	0.094
F	0.14	0.07	0.15	0.15	0.12	0.10	0.28	0.12	2.87	−0.63	0.127

* *p*: probability; D: duration; F: frequency; I: criteria; r: sum of the rows; y: sum of the columns.

**Table 4 ijerph-17-01977-t004:** Ranking results of alternatives.

Alternatives	D+	D−	CCi	Ranking
Curtilage (Alternative 1)	0.234	0.241	0.508	2
Neighborhood squares (Alternative 2)	0.203	0.171	0.457	4
Park (Alternative 3)	0.190	0.252	0.570	1
Campus (Alternative 4)	0.259	0.220	0.460	3

* D+: distance to weighted positive solution; D−: distance to weighted negative solution CCi: closeness coefficient.

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
