# Peer review of "Analysis of Environmental Activities for Developing Public Health Investments and Policies: A Comparative Study with Structure Equation and Interval Type 2 Fuzzy Hybrid Models"

_ijerph, 2020, doi:10.3390/ijerph17061977_

Round 1
Reviewer 1 Report
In this work the Authors are covering a very appealing theme of investigation.
- The structure is coherent, the design is sound as well as the methods employed so that the obtained results are logically backed and reported.
- In the Introduction, the theoretical framework would benefit by inserting an enlightening reference related to this article. (Manferdelli et al. Outdoor physical activity bears multiple benefits to health and society. J Sports Med Phys Fitness. 2019 May;59(5):868-879. doi: 10.23736/S0022-4707.18.08771-6. ).
Author Response
Istanbul, February 22th, 2020
Ivana Vujovic
Assistant Editor
Dear Ivana Vujovic,
According to your last email, please find attached a copy of the revised version of our paper, as a possible publication in the IJERPH:
Analysis of environmental activities for developing the public health investments and policies: A comparative study with structure equation and interval type 2 fuzzy hybrid models
The comments and suggestions for revision are reflected in the current version of the paper.
A response letter with main changes has been done in this revision together a separated individual response for each reviewer to their comments and a specific reference section to support our answers are attached.
Looking forward to hearing from you.
Thank you.
Yours sincerely,
Serhat Yüksel
Professor of Finance,
İstanbul Medipol University,
The School of Business
e-mail: serhatyuksel@medipol.edu.tr
Response letter to the Reviewers’ Comments on ijerph-715673
Title of the paper: Analysis of environmental activities for developing the public health investments and policies: A comparative study with structure equation and interval type 2 fuzzy hybrid models
First of all, authors would like to thank all the anonymous Reviewers and the Editors again for their efforts and valuable time to review and improve our paper.
Taking into account their constructive suggestions and comments, the paper has been carefully revised following the referees’ comments.
The main changes in the revised manuscript are:
- The introduction parts are improved.
- The literature review part is improved.
- The contribution of the study is underlined appropriately.
We provide below the responses to each referee’s comments.
Reviewer #1
Reviewer’s Comments: In this work the Authors are covering a very appealing theme of investigation.
- The structure is coherent, the design is sound as well as the methods employed so that the obtained results are logically backed and reported.
- In the Introduction, the theoretical framework would benefit by inserting an enlightening reference related to this article. (Manferdelli et al. Outdoor physical activity bears multiple benefits to health and society. J Sports Med Phys Fitness. 2019 May;59(5):868-879. doi: 10.23736/S0022-4707.18.08771-6. ).
Authors’ answers: First of all, we would like to thank the reviewer for his/her nice comments. According to the comments, we have added the following study in our reference list.
Maferdelli, G., La, A. T., & Codella, R. (2019). Outdoor physical activity bears multiple benefits to health and society. The Journal of sports medicine and physical fitness, 59(5), 868-879.

Reviewer 2 Report
This paper is very difficult to read, both for inadequate language and a completely chaotic structure. The introduction meanders through so many different topics that the reader is lost.
-> The introduction and literature review needs a complete rewrite focussing on physical activity in open spaces. No more, but also no less.
The massive formula explosion in the methods section is completely pointless to the reader and does not add to the understanding of the method. Please reframe this in more accessible terminology, and move the formulas to an appendix.
Moreover, please descriebe the the questionnaire in full detail, give examples of the items in the main text and the full questionnaire in the appendix. Discuss, why those items have been chosen, give an overview of the factor structure, fit measures etc. (You may refer to the APA's guidelines in reporting SEM, see attached, see also: Hoyle, R. H., & Isherwood, J. C. (2013). Reporting results from structural equation modeling analyses in Archives of Scientific Psychology. Archives of Scientific Psychology, 1, 14–22. http://dx.doi.org/10.1037/arc0000004).
Please try to be as concise as possible while being as narrative as needed. The article in its current form is exaggerating in concision (e.g. formulas and tables that are completely pointless to the reader), and very little narrative.
Improve the figures. The path diagram is too small; standardized coefficients would be nice (if this is possible/feasible with the four groups).
Improve the summary and conclusion by giving a clear take-home massage.

Author Response
Istanbul, February 22th, 2020
Ivana Vujovic
Assistant Editor
Dear Ivana Vujovic,
According to your last email, please find attached a copy of the revised version of our paper, as a possible publication in the IJERPH:
Analysis of environmental activities for developing the public health investments and policies: A comparative study with structure equation and interval type 2 fuzzy hybrid models
The comments and suggestions for revision are reflected in the current version of the paper.
A response letter with main changes has been done in this revision together a separated individual response for each reviewer to their comments and a specific reference section to support our answers are attached.
Looking forward to hearing from you.
Thank you.
Yours sincerely,
Serhat Yüksel
Professor of Finance,
İstanbul Medipol University,
The School of Business
e-mail: serhatyuksel@medipol.edu.tr
Response letter to the Reviewers’ Comments on ijerph-715673
Title of the paper: Analysis of environmental activities for developing the public health investments and policies: A comparative study with structure equation and interval type 2 fuzzy hybrid models
First of all, authors would like to thank all the anonymous Reviewers and the Editors again for their efforts and valuable time to review and improve our paper.
Taking into account their constructive suggestions and comments, the paper has been carefully revised following the referees’ comments.
The main changes in the revised manuscript are:
- The introduction parts are improved.
- The literature review part is improved.
- The contribution of the study is underlined appropriately.
We provide below the responses to each referee’s comments.
Reviewer #2
Reviewer’s Comments: The introduction and literature review needs a complete rewrite focussing on physical activity in open spaces. No more, but also no less.
Authors’ answers: First of all, we would like to thank the reviewer for his/her nice comments. According to the comments, we have made significant changes in the study. In this framework, we updated introduction and literature review parts. The introduction part is mainly designed by considering the issues related to only physical activity. This new introduction is given below.
“1. Introduction
Public health policies aim to maintain a healthy life and protect people from diseases [1–3]. There are a number of practices that are necessary to increase public health [4-6]. Physical activity of the public plays an important role for the improvement of public health. The main reason for this is that physical inactivity can cause some important health problems, such as obesity [7]. Therefore, people living in the country need to be encouraged to be more mobile. For this purpose, there are some things people can do [8–10]. For example, walking or cycling for transportation will help people become more active. In addition, performing fitness exercises and participating in sports activities are other practices that can minimize physical inactivity. These types of physical activity will contribute to the health of people. There are a number of strategies that can be implemented by countries to increase this issue. One of the most important issues in this process is the creation of environments within the country to support physical activities [11].
There are many different opportunities for people to have leisurely physical activity (LPA). In this context, there can be some facilities in which citizens can join some sport activities. For instance, wide open space for dancing and path for walking and jog can contribute LPA [12]. In addition to these issues, play equipment for kids and fitness equipment for adults are also other examples for this situation. Moreover, physical environment (PE) also plays a very key role for LPA [13]. Within this framework, the trees can be planted for shade, and fresh air and adequate trash bins can be provided to maintain sanitation. The main reason is that PE, such as noise and air have impact on physical activities. Additionally, amenities, such as tables, retail stores and toilets in the parks can promote LPA. Furthermore, aesthetics can also make contribution to higher LPA. For this purpose, green plants and water bodies can help people to make physical activity. In addition to these factors, maintenance and safety (MS) can have a positive influence on LPA as well [14,15].
In this context, it is very important to determine which physical activity will have more impact on public health. In other words, there are many different factors that improve public health. However, it may not be possible for policy makers to invest in all of them at the same time. It is possible to talk about many different reasons for this restriction [16]. Primarily, investing in the development of physical activities is very costly. Therefore, the state institution may not have sufficient budget to invest in all factors at the same time. Therefore, it is very important to identify and focus on priority investment areas in order to develop physical activities. In this way, it will be possible both to contribute to the development of physical activities and to use the public budget effectively [17,18].
Another important matter in this process is that the methodology used is effective. Otherwise, wrong results will be achieved, and this will lead to the implementation of wrong policies. Structural equation model is a new method preferred in order to examine the causality process between multiple variables especially in social sciences [19]. This method has been preferred for different purposes by different researchers, especially in recent years. It is thought that it will be effective in the emergence of the important issues in the analysis to be carried out in order to increase physical activities. In addition to the mentioned issue, multi-criteria decision-making methods are approaches used for this purpose in the literature. With methods such as AHP, ANP and DEMATEL, it is possible to identify the most important among the different alternatives [20-23]. These methods have also been taken into consideration within the framework of fuzzy logic in recent years. Considering the mentioned issues, it will be possible to determine which physical activities affect public health more with these approaches.
The allocation and user requirements of elements differ in various types of urban open space. This situation also leads to diverse effects of elements on LPA in different spaces. However, little research has been done on discrepancies of those influence. In this paper, it is aimed to evaluate environmental activities for developing the public health investments and policies Harbin in China. In this scope, literature is reviewed and 6 factors that influencing the environmental activities are defined. Additionally, duration and frequency are also taken into consideration. On the other side, 4 types of open spaces are also considered. For this purpose, two different analyses are performed. First of all, an evaluation is occurred by using a structural equation modelling. Moreover, IT2 fuzzy hybrid decision making model is proposed in the second analysis. In this framework, IT2 fuzzy DEMATEL and IT2 fuzzy TOPSIS methods are used.
The most important contribution of this study is performing two different analyses. With the help of this issue, a comparative analysis can be made, and this situation provides an opportunity to test the accuracy of the results. Hence, it is thought that the policy recommendations of this study can be more appropriate. In addition to this condition, structural equation modelling, IT2 fuzzy DEMATEL and IT2 fuzzy TOPSIS methods are used together firstly in this study regarding the evaluation of environmental activities for developing the public health investments and policies. Moreover, the results of this study will be leading both policy makers and academicians. In other words, by considering the analysis results, it can be much easier to improve public health with the help of correct policies with respect to the environmental activities.
This study consists of 5 different sections. In the first part, it is stated why issues such as physical activity and public health are important. Similar studies in the literature are summarized in the second part of the study. In addition, the third part of the study provides information about the methods used in the analysis. In the fourth part of the study, the details of the two different analyzes are given. In this framework, both the structural equation model and the analysis made with IT2 fuzzy multi-criteria decision-making methods are shared. The last part includes of discussion and conclusion.
Additionally, the literature review part is also updated. In this context, the studies only related to the physical activities are included in this part. Because of this situation, we have to delete many studies in this part since they are not directly related to the physical activity. Additionally, we have also added some studies in this part in order to improve the quality of literature review. New literature review becomes as following
“2. Literature Review
The issue that environmental factors are also effective on public health has been discussed in the literature by many researchers. In this process, one of the most significant topics discussed by the researchers is the physical activity. Within this framework, some researchers argued that physical activities can be improved when there is low environmental pollution. For this situation, Ogasawara et al. [12] and Finch et al. [24] identified that the prevention of hazardous waste and minimizing environmental pollution leads to higher physical activities and this issue also contributes to the improvement of public health. Similarly, Armand et al. [13] and Lu et al. [25] focused on the relationship between environmental pollution, economic development and public health. It is determined that if air and water are not polluted, it can be more possible to increase physical activities which are prominent issues for citizens to lead a healthy life.
In addition to them, some researchers in the literature also defined that green areas are very important to increase the physical activities of the citizens. For this situation, it is mainly discussed that in the country, there should be lots of green areas for the people to make these activities. In this framework, McLafferty and Murray [14] and Moretti et al. [26] aimed to evaluate the environmental and human health impact of road construction activities. They mainly stated that physical activities will be adversely affected if the green areas are destroyed and the construction is done. Parallel to these studies, Henry and Price [15] and Pilkington et al. [27] also focused on the effects on environmental factors on the physical activities. In this study, it is identified that if environments are not well designed, there is a risk that physical activities in this country can be lowered.
Moreover, Hunter et al. [16] and Cronk and Bartram [28] also tried to examine the environmental conditions in health care facilities in low-and middle-income countries. According to the results of the analysis, it is identified that in order to increase public health, citizens should have areas to perform physical activity. In this context, environmental factors play a very key role for the people to join physical activities. In other words, if there are lots of green areas in the country, this situation attracts the attention of the people to make these activities. Chauvin et al. [17] and Jang et al. [29] made also similar study for this issue. They also underlined the importance of green areas to improve physical activities in the country. In addition to these studies, Kojan et al. [19] and Morici et al. [30] also made an analysis to understand the relationship between environmental issues and physical activities. They reached the conclusion that measures should be taken to reduce environmental pollution in order to improve physical activities. Hence, it can be much easier to implement an effective public health policy.
In the literature, some researchers also identified the main advantages of the physical activities in improving public health. It is observed that the public can prevent many diseases by doing physical activities. For example, Dudley et al. [31] made an evaluation regarding the physical literacy policy in public health. In this study, it is mainly concluded that physical activities play a very significant role to improve public health. Therefore, it has been stated that the public should have environments that can perform physical activity. Additionally, White et al. [32] also aimed to examine physical activity in natural environments to have better health condition. In this context, it is recommended to put physical activity equipment in places such as the park that everyone can reach.
In addition, some studies in the literature determined that the creation of areas such as walking trails, where the public will engage in physical activity, will also contribute to the development of this issue. Hulteen et al. [33] aimed to understand how global participation in sport and leisure-time physical activities can be increased. They argued that necessary areas should be created for the people to make physical activities. Parallel to this study, Khan et al. [34] also focused on the patterns of physical activity in adolescents in Dhaka city of Bangladesh. They mainly stated that for a more effective public health policy, the public should be ensured to participate in physical activities. Kobau et al. [35] and Saunders et al. [36] also determined that as a result of physical activities, some diseases will decrease, and this will contribute to the improvement of public health.
As a result of the literature review, it was determined that many different analyzes were made on the subject of public health. In the studies, issues such as the importance of public health, how it can be increased and what it has a positive effect on were examined. In these studies, it is seen that regression analysis and survey methods are generally preferred. However, it is thought that more specific suggestions are needed to increase public health, and studies are needed. In this study, it was focused on physical activity to increase public health. In this framework, a study has been conducted on what kinds of physical activity opportunities should be created. Therefore, it is thought that this detailed study will be directed towards this need stated in the literature.
Reviewer’s Comments: The massive formula explosion in the methods section is completely pointless to the reader and does not add to the understanding of the method. Please reframe this in more accessible terminology, and move the formulas to an appendix.
Authors’ answers: According to the comments, we have made significant changes in the study. In this framework, all formulas are transferred to the appendix part. Additionally, most of the tables are also copied in the appendix. Also, we have made a detailed explanation in the methodology. Now, it is thought that it becomes much easier to follow for the readers. Moreover, we have also improved the discussion and conclusion part. For this purpose, we have defined our results more effectively. The details of this part are demonstrated below.
“This paper tries to evaluate environmental activities for developing the public health investments and policies Harbin in China. Within this framework, as a result of the literature review, 6 factors that influencing the environmental activities are identified. Moreover, duration and frequency are also considered. Furthermore, 4 types of open spaces are also taken into consideration. In this scope, two different analyses are conducted. Firstly, an evaluation is occurred by using a structural equation modelling. Additionally, IT2 fuzzy hybrid decision making model is proposed in the second analysis. Within this context, IT2 fuzzy DEMATEL and IT2 fuzzy TOPSIS methods are used.
According to the analysis results, it is determined that facilities play the most significant role to improve LPA. In addition to this situation, it is also concluded that maintenance and safety is also another important criterion to increase these activities. However, it is identified that accessibility and amenities are the least important items for this purpose. Another important conclusion of this study is that both SEM and IT2 fuzzy logic analysis results are quite coherent. Hence, it is recommended that some sport activities, such as wide-open space for dancing and path for walking and jog can contribute the improvement of LPA. On the other side, it is also learnt that people, who want to join physical activities, should prefer to feel secured. Thus, governmental authorities should take necessary actions for the people to feel themselves secured while joining these activities.”
According to the comments, we have improved the discussion part. Within this context, we have also detailed the limitations of the study. Now it becomes as follows.
“The main limitation of this study is making evaluation only for Harbin. Therefore, in a new study, an analysis can be conducted for other regions or countries. With the help of this issue, different recommendations can be presented. In addition to this condition, another important limitation of this study is related to the methodology. Hence, it is recommended that different methods can also be considered. For instance, probit, logit and regression analyses can be conducted. Hence, it can be possible to compare the results of different studies. Moreover, it can also be accepted as the limitation of this study that only the concept of physical activities is evaluated. In addition to the physical activity, there are also lots of other factors which have a contribution to the public health, such as economic growth and green environment. In the future studies, these factors can be taken into account.”
Reviewer’s Comments: Moreover, please descriebe the the questionnaire in full detail, give examples of the items in the main text and the full questionnaire in the appendix. Discuss, why those items have been chosen, give an overview of the factor structure, fit measures etc. (You may refer to the APA's guidelines in reporting SEM, see attached, see also: Hoyle, R. H., & Isherwood, J. C. (2013). Reporting results from structural equation modeling analyses in Archives of Scientific Psychology. Archives of Scientific Psychology, 1, 14–22. http://dx.doi.org/10.1037/arc0000004).
Authors’ answers: Based on the reviewer’s comments, we have described the full questionnaire in the appendix. On the other side, in the main text, we have also discussed why those ítems are chosen. The related part in the main text are given below.
“In this study, two different analyses are performed. In the first stage, Structure Equation Model (SEM) is applied for understanding the behavioral results more accurately. This model is one of the most important statistical method in the field of behavioral and social sciences. Simultaneous analysis of several groups (SASG) can be used to analyze whether the theoretical model proposed by the researchers is the same in different groups. It can also test the discrepancies of related paths in the same group. In this paper, the discrepancies of the relationships in 4 types of open spaces is studied by SASG. Randomly selected respondents are administered a questionnaire survey. Additionally, three pieces of information included the quality evaluation of the elements, self-rated LPA (the frequency every week and duration every time) and users’ socioeconomic background are collected. The evaluation of elements is measured by five-points Likert scale, which are very poor, poor, average, good and very good. Elements include 6 different domains {facilities, accessibility, physical environment (PE), amenities, aesthetics and maintenance & safety (MS)} and 42 items involved in the questionnaire are derived from the previous studies. Furthermore, 400 questionnaires are randomly distributed in 8 urban open spaces in Harbin, included 4 space types (curtilage, NS, park, campus). The survey recovers 322 valid questionnaires with a recovery rate of 80.5%.”
Additionally, the full questionnaire in the appendix part is described below.
Part 2:What is your evaluation of the quality of the elements in the site?(Please tick √ on your option)
Elements |
Five-points Likert scale |
||||
Facilities |
very poor |
poor |
average |
good |
very good |
1.play equipment |
|
|
|
|
|
2.anti-disturbing installation |
|
|
|
|
|
3.protection facilities |
|
|
|
|
|
4.equipment for fitness |
|
|
|
|
|
5.instructions of equipment |
|
|
|
|
|
6.square |
|
|
|
|
|
7.size |
|
|
|
|
|
8.ping-pong table |
|
|
|
|
|
9.badminton area |
|
|
|
|
|
10.court |
|
|
|
|
|
11.paths for walking |
|
|
|
|
|
12.paths for riding |
|
|
|
|
|
13.seating |
|
|
|
|
|
14.the material of seat |
|
|
|
|
|
Accessibility |
very poor |
poor |
average |
good |
very good |
15.walkable |
|
|
|
|
|
16.proximity |
|
|
|
|
|
17.enjoyable street |
|
|
|
|
|
Physical Environment (PE) |
very poor |
poor |
average |
good |
very good |
18.noise |
|
|
|
|
|
19.fresh air |
|
|
|
|
|
20.shade |
|
|
|
|
|
21.leeward |
|
|
|
|
|
Amenities |
very poor |
poor |
average |
good |
very good |
22.toilets |
|
|
|
|
|
23.bulletin board |
|
|
|
|
|
24.chessboard |
|
|
|
|
|
25.tables |
|
|
|
|
|
26.drink fountains |
|
|
|
|
|
27.retail stores |
|
|
|
|
|
28.bins |
|
|
|
|
|
Aesthetics |
very poor |
poor |
average |
good |
very good |
29.nature |
|
|
|
|
|
30.artificial landscape |
|
|
|
|
|
31.greenery |
|
|
|
|
|
32.building elements |
|
|
|
|
|
33.winter landscape |
|
|
|
|
|
Maintenance & Safety (MS) |
very poor |
poor |
average |
good |
very good |
34.amenities maintenance |
|
|
|
|
|
35.equipment maintenance |
|
|
|
|
|
36.cleaning |
|
|
|
|
|
37.snow removal |
|
|
|
|
|
38.lighting |
|
|
|
|
|
39.surface |
|
|
|
|
|
40.security |
|
|
|
|
|
41.equipment safety |
|
|
|
|
|
42.signature |
|
|
|
|
|
Reviewer’s Comments: Please try to be as concise as possible while being as narrative as needed. The article in its current form is exaggerating in concision (e.g. formulas and tables that are completely pointless to the reader), and very little narrative.
Authors’ answers: According to the comments, we have made significant changes in the study. In this framework, all formulas are transferred to the appendix part. Additionally, most of the tables are also copied in the appendix. Also, we have made a detailed explanation in the methodology. Now, it is thought that it becomes much easier to follow for the readers.
Reviewer’s Comments: Improve the figures. The path diagram is too small; standardized coefficients would be nice (if this is possible/feasible with the four groups).
Authors’ answers: Based on the reviewer comments, we have improved the quality of the figures.
Reviewer’s Comments: Improve the summary and conclusion by giving a clear take-home massage.
Authors’ answers: We have improved the discussion and conclusion part. For this purpose, we have defined our results more effectively. The details of this part are demonstrated below.
“This paper tries to evaluate environmental activities for developing the public health investments and policies Harbin in China. Within this framework, as a result of the literature review, 6 factors that influencing the environmental activities are identified. Moreover, duration and frequency are also considered. Furthermore, 4 types of open spaces are also taken into consideration. In this scope, two different analyses are conducted. Firstly, an evaluation is occurred by using a structural equation modelling. Additionally, IT2 fuzzy hybrid decision making model is proposed in the second analysis. Within this context, IT2 fuzzy DEMATEL and IT2 fuzzy TOPSIS methods are used.
According to the analysis results, it is determined that facilities play the most significant role to improve LPA. In addition to this situation, it is also concluded that maintenance and safety is also another important criterion to increase these activities. However, it is identified that accessibility and amenities are the least important items for this purpose. Another important conclusion of this study is that both SEM and IT2 fuzzy logic analysis results are quite coherent. Hence, it is recommended that some sport activities, such as wide-open space for dancing and path for walking and jog can contribute the improvement of LPA. On the other side, it is also learnt that people, who want to join physical activities, should prefer to feel secured. Thus, governmental authorities should take necessary actions for the people to feel themselves secured while joining these activities.”
According to the comments, we have improved the discussion part. Within this context, we have also detailed the limitations of the study. Now it becomes as follows.
“The main limitation of this study is making evaluation only for Harbin. Therefore, in a new study, an analysis can be conducted for other regions or countries. With the help of this issue, different recommendations can be presented. In addition to this condition, another important limitation of this study is related to the methodology. Hence, it is recommended that different methods can also be considered. For instance, probit, logit and regression analyses can be conducted. Hence, it can be possible to compare the results of different studies. Moreover, it can also be accepted as the limitation of this study that only the concept of physical activities is evaluated. In addition to the physical activity, there are also lots of other factors which have a contribution to the public health, such as economic growth and green environment. In the future studies, these factors can be taken into account.”

Reviewer 3 Report
Your paper is very interesting, and based on a very complex methodology, providing very complex and "hard to be read" results. I really don't know the methodology that you used, and I don't think that many of the readers know it. I would suggest you to :
explain in a better way the methodology, I read it again and I still don't understand what has been done link in a better way the results and the discussion, explaining in which way your methodology can help to get those results, and thus suggest that policies starting from the results.
It sounds strange when you write that the only limit of your research is the fact that it has been done only in Herbin. It's a little been pretentious...
Author Response
Istanbul, February 22th, 2020
Ivana Vujovic
Assistant Editor
Dear Ivana Vujovic,
According to your last email, please find attached a copy of the revised version of our paper, as a possible publication in the IJERPH:
Analysis of environmental activities for developing the public health investments and policies: A comparative study with structure equation and interval type 2 fuzzy hybrid models
The comments and suggestions for revision are reflected in the current version of the paper.
A response letter with main changes has been done in this revision together a separated individual response for each reviewer to their comments and a specific reference section to support our answers are attached.
Looking forward to hearing from you.
Thank you.
Yours sincerely,
Serhat Yüksel
Professor of Finance,
İstanbul Medipol University,
The School of Business
e-mail: serhatyuksel@medipol.edu.tr
Response letter to the Reviewers’ Comments on ijerph-715673
Title of the paper: Analysis of environmental activities for developing the public health investments and policies: A comparative study with structure equation and interval type 2 fuzzy hybrid models
First of all, authors would like to thank all the anonymous Reviewers and the Editors again for their efforts and valuable time to review and improve our paper.
Taking into account their constructive suggestions and comments, the paper has been carefully revised following the referees’ comments.
The main changes in the revised manuscript are:
- The introduction parts are improved.
- The literature review part is improved.
- The contribution of the study is underlined appropriately.
We provide below the responses to each referee’s comments.
Reviewer #3
Reviewer’s Comments: Your paper is very interesting, and based on a very complex methodology, providing very complex and "hard to be read" results. I really don't know the methodology that you used, and I don't think that many of the readers know it. I would suggest you to :
explain in a better way the methodology, I read it again and I still don't understand what has been done link in a better way the results and the discussion, explaining in which way your methodology can help to get those results, and thus suggest that policies starting from the results.
Authors’ answers: First of all, we would like to thank the reviewer for his/her nice comments. According to the comments, we have made significant changes in the study. In this framework, all formulas are transferred to the appendix part. Additionally, most of the tables are also copied in the appendix. Also, we have made a detailed explanation in the methodology. Now, it is thought that it becomes much easier to follow for the readers. Moreover, we have also improved the discussion and conclusion part. For this purpose, we have defined our results more effectively. The details of this part are demonstrated below.
“This paper tries to evaluate environmental activities for developing the public health investments and policies Harbin in China. Within this framework, as a result of the literature review, 6 factors that influencing the environmental activities are identified. Moreover, duration and frequency are also considered. Furthermore, 4 types of open spaces are also taken into consideration. In this scope, two different analyses are conducted. Firstly, an evaluation is occurred by using a structural equation modelling. Additionally, IT2 fuzzy hybrid decision making model is proposed in the second analysis. Within this context, IT2 fuzzy DEMATEL and IT2 fuzzy TOPSIS methods are used.
According to the analysis results, it is determined that facilities play the most significant role to improve LPA. In addition to this situation, it is also concluded that maintenance and safety is also another important criterion to increase these activities. However, it is identified that accessibility and amenities are the least important items for this purpose. Another important conclusion of this study is that both SEM and IT2 fuzzy logic analysis results are quite coherent. Hence, it is recommended that some sport activities, such as wide-open space for dancing and path for walking and jog can contribute the improvement of LPA. On the other side, it is also learnt that people, who want to join physical activities, should prefer to feel secured. Thus, governmental authorities should take necessary actions for the people to feel themselves secured while joining these activities.”
Reviewer’s Comments: It sounds strange when you write that the only limit of your research is the fact that it has been done only in Herbin. It's a little been pretentious...
Authors’ answers: According to the comments, we have improved the discussion part. Within this context, we have also detailed the limitations of the study. Now it becomes as follows.
“The main limitation of this study is making evaluation only for Harbin. Therefore, in a new study, an analysis can be conducted for other regions or countries. With the help of this issue, different recommendations can be presented. In addition to this condition, another important limitation of this study is related to the methodology. Hence, it is recommended that different methods can also be considered. For instance, probit, logit and regression analyses can be conducted. Hence, it can be possible to compare the results of different studies. Moreover, it can also be accepted as the limitation of this study that only the concept of physical activities is evaluated. In addition to the physical activity, there are also lots of other factors which have a contribution to the public health, such as economic growth and green environment. In the future studies, these factors can be taken into account.”

Round 2
Reviewer 2 Report
Thank you for the revised manuscript. It has much improved with respect to readability. However, to my understanding, there is some work to do:
- Specific considerations on methodology do not belong to the introduction (especially the section line 78ff).
Sentences like"Another important matter in this process is that the methodology used is effective. Otherwise, wrong results will be achieved, and this will lead to the implementation of wrong policies."
are trivial and should be avoided. Please carefully scan the document for such "no-information-sentences". And it is not exactly true that SEM is a new method. It has been around since the 1960ies with roots in the 1920ies... (Wright's path analysis); Again, please avoid such null-information. - All abbreviations need to be explained and introduced before used (AHP, ANP, DEMATEL)
- I do not see the point in presenting "3. Methodology" and parts of the methodology in "4. Analysis" (which is in fact the "results" chapter); moreover, chapter (3) is comprised of one (!) paragraph (13 lines...), which is in vast misbalance to the other chapters;
In order to make (3) more round, move the parts from the introduction here, and describe in gentle words both the approach and the methodology based on a research question and/or hypotheses that should be clearly stated at the end of chapter (2).
Moreover, all information regarding the sampling, choice of research design and location etc. should be presented here, and the chapter titled "3. Materials and Methods". - Rename "4. Analysis" with "4. Results"; move the information on the questionnaire and the sampling (lines 194ff) to the previous chapter.
- Segment the Results chapter with two sub-chapters, one for each analytical approach.
- Figure 1 (SEM) needs a caption explaining the main results. Please mark significant paths; please make sure that the coefficients do not lie on path lines or too close together (readability).
- Tables1-4 and A1-12 need a captions explaining the main results and the abbreviations used.
- The conclusion still lacks a clear take-home message;
- You should not end the article with a limitation; place limitations before the conclusion;
- Limitations: what would be the expected difference with "probit, logic and regression analyses" to SEM the machine-learning approach you have taken?
Author Response
Istanbul, March 4th, 2020
Ivana Vujovic
Assistant Editor
Dear Ivana Vujovic,
According to your last email, please find attached a copy of the revised version of our paper, as a possible publication in the IJERPH:
Analysis of environmental activities for developing the public health investments and policies: A comparative study with structure equation and interval type 2 fuzzy hybrid models
The comments and suggestions for revision are reflected in the current version of the paper.
A response letter with main changes has been done in this revision together a separated individual response for each reviewer to their comments and a specific reference section to support our answers are attached.
Looking forward to hearing from you.
Thank you.
Yours sincerely,
Serhat Yüksel
Professor of Finance,
İstanbul Medipol University,
The School of Business
e-mail: serhatyuksel@medipol.edu.tr
Response letter to the Reviewers’ Comments on ijerph-715673
Title of the paper: Analysis of environmental activities for developing the public health investments and policies: A comparative study with structure equation and interval type 2 fuzzy hybrid models
First of all, authors would like to thank all the anonymous Reviewers and the Editors again for their efforts and valuable time to review and improve our paper.
Taking into account their constructive suggestions and comments, the paper has been carefully revised following the referees’ comments.
The main changes in the revised manuscript are:
- The introduction parts are improved.
- The literature review part is improved.
- The contribution of the study is underlined appropriately.
We provide below the responses to each referee’s comments.
Reviewer #2
Reviewer’s Comments: Specific considerations on methodology do not belong to the introduction (especially the section line 78ff).
Sentences like
"Another important matter in this process is that the methodology used is effective. Otherwise, wrong results will be achieved, and this will lead to the implementation of wrong policies."
are trivial and should be avoided. Please carefully scan the document for such "no-information-sentences". And it is not exactly true that SEM is a new method. It has been around since the 1960ies with roots in the 1920ies... (Wright's path analysis); Again, please avoid such null-information.
Authors’ answers: We have updated our paper based on the reviewer’s comments. In this framework, these specific considerations are eliminated from the study. New paragraph became as following.
“Structural equation model is one of well-known methods in order to examine the causality process between multiple variables especially in social sciences [19]. This method has been preferred for different purposes by different researchers, especially in recent years. It is thought that it will be effective in the emergence of the important issues in the analysis to be carried out in order to increase physical activities. In addition to the mentioned issue, multi-criteria decision-making methods are approaches used for this purpose in the literature. With methods such as AHP, ANP and DEMATEL, it is possible to identify the most important among the different alternatives [20-23]. These methods have also been taken into consideration within the framework of fuzzy logic in recent years. Considering the mentioned issues, it will be possible to determine which physical activities affect public health more with these approaches.”
Reviewer’s Comments: 2. All abbreviations need to be explained and introduced before used (AHP, ANP, DEMATEL)
Authors’ answers: All abbreviations are explained in full version before they are used.
Reviewer’s Comments: 3. I do not see the point in presenting "3. Methodology" and parts of the methodology in "4. Analysis" (which is in fact the "results" chapter); moreover, chapter (3) is comprised of one (!) paragraph (13 lines...), which is in vast misbalance to the other chapters;
In order to make (3) more round, move the parts from the introduction here, and describe in gentle words both the approach and the methodology based on a research question and/or hypotheses that should be clearly stated at the end of chapter (2).
Moreover, all information regarding the sampling, choice of research design and location etc. should be presented here, and the chapter titled "3. Materials and Methods".
Authors’ answers: Based on the reviewer’s comments, we have improved the article. In this scope, the title is changed as 3. Materials and Methods". Under this title, firstly, the hypthesis of the studiy is explained. After that, the theoretical parts are transferred to this section. New versión is given below.
“3. Materials and Methods
In this paper, it is aimed to examine the environmental activities for developing the public health investments and policies Harbin in China. For this purpose, two different analyses are performed. First of all, an evaluation is occurred by using a structural equation modeling. Moreover, IT2 fuzzy hybrid decision making model is proposed in the second analysis. In this framework, IT2 fuzzy DEMATEL and IT2 fuzzy TOPSIS methods are used. IT2 fuzzy sets aim to minimize uncertainty of interval type-1 fuzzy sets. For this purpose, upper and lower trapezoidal membership functions are considered [37,38]. The details of calculations of IT2 fuzzy sets are given in the appendix part. DEMATEL approach is considered to find the significance values of different alternatives. It is also possible to make impact relationship analysis with this methodology. Thus, it is thought that DEMATEL has some advantages over the similar methodologies [39]. For instance, it is possible to generate impact relation map between the criteria. In this process, initial direct-relation fuzzy matrix, normalized matrix and total relation matrix are created so that the weights of the factors can be defined. The details of this process are also shared in the appendix part as well. On the other side, TOPSIS methodology is also used to rank different alternatives according to their importance. This approach is mainly considered to understand which alternatives are more effective in comparison with others [41]. Its calculation process is also explained in the appendix part.
Moreover, a comparative analysis is applied for ranking the alternatives of environmental activities to develop the most appropriate public health policies. For this purpose, 6 factors that influencing the environmental activities are defined based on the literature review and they are presented in Table 1.
Table 1. Factors Influencing the Environmental Activities.
Factors |
Supported Literature |
facilities (I1) |
Cron andBartram [28]; Jang et al. [29] |
accessibility (I2) |
Morici et al. [30]; White et al. [32] |
physical environment (I3) |
Hulteen et al. [33]; Khan et al. [34] |
amenities (I4) |
Jang et al. [29]; Saunders et al. [36] |
aesthetics (I5) |
Finch et al. [24]; Lu et al. [25] |
maintenance & safety (I6) |
Chauvin et al. [17]; Kojan et al. [18] |
As can be seen from Table 1, 6 different environmental factors were identified in order to increase physical activity. First of all, it is possible to increase the participation of the people in physical activities with the help of some facilities. In this framework, walking track and exercise equipment contribute to achieving this goal. On the other hand, these facilities must be at an accessible location in order to increase physical activities. Otherwise, it will not be possible to sustain these physical activities. In addition, the quality of the physical environment is important in increasing physical activities. In this context, it is an important advantage to find forest areas in the living environment. In addition, the fact that the air is not dirty plays an important role in this framework. Moreover, amenities, such as tables, retail stores and toilets in the parks can contribute the improvement of the increase physical activities. Also, aesthetics like green plants and water bodies can help people to make physical activities. Additionally, people, who want to join physical activities, should prefer to feel secured. Different environmental activities for the public health are determined to analyze the leisurely public activities. For this purpose, 4 alternatives of leisurely public activities are selected as curtilage (alternative 1), neighborhood square (alternative 2), park (alternative 3), and campus (alternative 4). The evaluations are provided by considering the environmental activities of Harbin in China.
In this study, two different analyses are performed. In the first stage, Structure Equation Model (SEM) is applied for understanding the behavioral results more accurately. This model is one of the most important statistical method in the field of behavioral and social sciences. Simultaneous analysis of several groups (SASG) can be used to analyze whether the theoretical model proposed by the researchers is the same in different groups. It can also test the discrepancies of related paths in the same group. In this paper, the discrepancies of the relationships in 4 types of open spaces is studied by SASG. Randomly selected respondents are administered a questionnaire survey. Additionally, three pieces of information included the quality evaluation of the elements, self-rated LPA (the frequency every week and duration every time) and users’ socioeconomic background are collected. The evaluation of elements is measured by five-points Likert scale, which are very poor, poor, average, good and very good. Elements include 6 different domains {facilities, accessibility, physical environment (PE), amenities, aesthetics and maintenance & safety (MS)} and 42 items involved in the questionnaire are derived from the previous studies. Furthermore, 400 questionnaires are randomly distributed in 8 urban open spaces in Harbin, included 4 space types (curtilage, NS, park, campus). The survey recovers 322 valid questionnaires with a recovery rate of 80.5%.
In the second stage, a hybrid fuzzy decision-making model based on IT2 fuzzy sets is used for comparing the analysis results of factors and environmental activities. 3 decision makers that are experts in the field of environment and public health are appointed for the linguistic evaluations for the factors and physical activities. IT2 FDEMATEL is applied for weighting the factors of environmental activities and IT2 FTOPSIS is used for ranking the alternatives of leisurely physical activities. Thus, it is possible to understand the coherency of analysis and to develop the public health policies in detail. In this study, two different analyses are performed. With respect to the structure equation model results, a survey study is made with 322 different respondents. The details of the survey questions and these people are given on the appendix part (Table A1).”
Reviewer’s Comments: 4.Rename "4. Analysis" with "4. Results"; move the information on the questionnaire and the sampling (lines 194ff) to the previous chapter.
5.Segment the Results chapter with two sub-chapters, one for each analytical approach.
Authors’ answers: According to the comments, the name of this section is changed by 4. Results. Under this title, we have created two different subtitles according to each analytical approach. New version is given below.
“4. Results
4.1. Analysis Results of Structural Equation Method
4.2. Analysis Results of Fuzzy MCDM Approaches “
Reviewer’s Comments: Figure 1 (SEM) needs a caption explaining the main results. Please mark significant paths; please make sure that the coefficients do not lie on path lines or too close together (readability).
Authors’ answers: We have increased the size of the Figure 1 and it is believed that now it becomes more readable.
Reviewer’s Comments: 7.Tables1-4 and A1-12 need a captions explaining the main results and the abbreviations used.
Authors’ answers: We have explained all terms of these tables at bottom.
Reviewer’s Comments: The conclusion still lacks a clear take-home message;
Authors’ answers: We have improved this part. For this purpose, a clear take-home message is given. In other words, to reach the conclusion, more specific issues are generated. In addition to this situation, a comparison is also made with similar studies in the literature.
Reviewer’s Comments: You should not end the article with a limitation; place limitations before the conclusion;
Authors’ answers: The limitation paragraph is put before conclusion.
Reviewer’s Comments: Limitations: what would be the expected difference with "probit, logic and regression analyses" to SEM the machine-learning approach you have taken?
Authors’ answers: The main contribution of using these methods in the future studies is explained in the limitation paragraph. The new version of the paragraph is given below.
“The main limitation of this study is making evaluation only for Harbin. Therefore, in a new study, an analysis can be conducted for other regions or countries. With the help of this issue, different recommendations can be presented. In addition to this condition, another important limitation of this study is related to the methodology. Hence, it is recommended that different methods can also be considered. For instance, probit, logit and regression analyses can be conducted. Hence, it can be possible to compare the results of different studies. The main reason is that probit, logit and regression analyses consider quantitative data whereas SEM and fuzzy MCDM approaches use the opinions of people and experts. Thus, it can be possible to make comparative analysis. Moreover, it can also be accepted as the limitation of this study that only the concept of physical activities is evaluated. In addition to the physical activity, there are also lots of other factors which have a contribution to the public health, such as economic growth and green environment. In the future studies, these factors can be taken into account.”